# Recruiters' perspectives and experiences of trial recruitment processes: a qualitative evidence synthesis protocol

Nicola Farrar ,[1] Daisy Elliott ,[1] Marcus Jepson ,[1] Catherine Houghton ,[2] Bridget Young ,[3] Jenny Donovan ,[1] Leila Rooshenas [1]

¹Population Health Sciences, University of Bristol, Bristol, UK
²School of Nursing and Midwifery, National University of Ireland Galway, Galway, Galway, Ireland
³Department of Health Services Research, University of Liverpool, Liverpool, UK

**Correspondence to**
Nicola Farrar;
nf0196@bristol.ac.uk

## ABSTRACT

**Introduction** Recruitment to randomised trials (RCTs) is often challenging. Reviews of interventions to improve recruitment have highlighted a paucity of effective interventions aimed at recruiters and the need for further research in this area. Understanding the perspectives and experiences of those involved in RCT recruitment can help to identify barriers and facilitators to recruitment, and subsequently inform future interventions to support recruitment. This protocol describes methods for a proposed qualitative evidence synthesis (QES) of recruiters' perspectives and experiences relating to RCT recruitment.

**Methods and analysis** The proposed review will synthesise studies reporting clinical and non-clinical recruiters' perspectives and experiences of recruiting to RCTs. The following databases will be searched: Ovid MEDLINE, CINAHL, EMBASE, PsycInfo, Cochrane Central Register of Controlled Trials, ORRCA and Web of Science. A thematic synthesis approach to analysing the data will be used. An assessment of methodological limitations of each study will be performed using the Critical Appraisal Skills Programme tool. Assessing the confidence in the review findings will be evaluated using the GRADE Confidence in Evidence from Reviews of Qualitative research (GRADE-CERQual) tool.

**Ethics and dissemination** The proposed QES will not require ethical approval as it includes only published literature. The results of the synthesis will be published in a peer-reviewed journal and publicised using social media. The results will be considered alongside other work addressing factors affecting recruitment in order to inform future development and refinement of recruitment interventions.

**PROSPERO registration number** CRD42020141297.

## INTRODUCTION

Randomised controlled trials (RCTs) are considered the gold standard method for producing evidence of effectiveness of interventions,[1] but a recent review found that only 56% of publicly funded (National Institute for Health Research Health Technology Assessment) RCTs were able to recruit their final target sample size.[2] This can have resource implications, as trials with slower than anticipated recruitment may request extra funding

## Strengths and limitations of this study

► This protocol outlines a detailed account of plans for undertaking a forthcoming qualitative evidence synthesis on recruiters' perspectives and experiences relating to randomised controlled trials recruitment, something which has been called for in the literature.
► The protocol has been developed by a group of researchers with a range of disciplinary backgrounds and levels of experience.
► The proposed synthesis will use a thematic approach to synthesising the data, as endorsed by Thomas and Harden.
► The proposed synthesis will use established and Cochrane endorsed methods of assessing the confidence in the findings produced by the review, using the GRADE Confidence in the Evidence from Reviews of Qualitative Research approach.
► The proposed synthesis will be limited to including only English language texts.

for extensions, or closure prematurely.[3] Poor recruitment can also delay the reporting of findings that could potentially change patient care. As such improving recruitment has been identified as an important priority for trials methodology research.[4] Recruitment is a complex process which can be influenced by several factors including the patients and professionals undertaking recruitment ('recruiters').[5 6] Many studies exploring this issue have highlighted the patient-related factors[7–9] or practical considerations such as availability of staff and resources to support recruitment.[10]

The literature on patient-related factors is well developed, with a Cochrane evidence synthesis recently reporting its findings.[11] The literature on recruiter perspectives and experiences of recruitment-lacks an up-to-date synthesis.[11] One review, published in 2012, used systematic review methods to identify strategies aimed at improving clinicians' recruitment activity in RCTs and qualitative

methods to synthesise clinicians' attitudes towards RCT recruitment.[5] The authors identified that there were methodological challenges associated with the broad scope of their review, but concluded that using qualitative methods to understand and overcome barriers to recruitment was a promising intervention, and that future interventions to improve recruitment could also target understanding and communication of RCT methods.[5] Other research published in 2014 addressed the broader topic of clinicians' and researchers' perspectives of recruiting to clinical research,[12] rather than specifically RCTs, and therefore may not fully reflect the experiences of recruiters that are unique to RCT recruitment. In an early version of Treweek *et al*'s Cochrane systematic review on methods to improve recruitment to RCTs, the authors identified that research on interventions to improve recruitment targeting recruiters was lacking, and may benefit from further investment,[3] with little changing in the 2018 update.[13]

Since the publication of the above reviews a growing body of research has been published, including new insights into the emotional and intellectual challenges recruiters can encounter while making decisions about approaching eligible patients[6 14] and explaining the trial,[15] and the need for training and support for recruiters to overcome these. A up-to-date evidence synthesis of recruiters' (that encompasses all professional roles) perspectives and experiences of recruiting to RCTs (specifically) is now warranted,[11] given the limitations of previous reviews of recruiter perspectives, and the growth of interest on this topic since the last search (2013) was conducted. Such a review, combined with the 2020 Cochrane review on patient perspectives,[11] and existing reviews on recruitment intervention effectiveness,[13 16] will enable a comprehensive understanding of the myriad of factors that can support or hinder RCT recruitment. This will help to inform the design of relevant recruitment interventions and illuminate areas in need of future primary research.

### Objectives
This protocol describes a qualitative evidence synthesis (QES) that will identify, appraise and synthesise existing evidence regarding the perspectives and experiences of recruitment staff who actively approach potential participants in a healthcare setting to take part in RCTs.

### METHODS AND ANALYSIS
### Criteria for considering studies for the synthesis
The SPIDER (Sample, Phenomenon of Interest, Design, Evaluation, Research type) search tool is a recognised as an appropriate tool in QES and was used to define the research question and search terms.[17] It was modified to include an additional heading of 'comparison' as is more commonly used when applying the popular PICO (population, intervention, comparison, outcome) tool[17] in quantitative or mixed-methods research.

### Search methods for identification of studies
Retrieval of qualitative literature can be challenging.[18] Booth[19] notes the popularity of the technique of using 'building blocks' (p.314) to define search terms. Initial search terms have been compiled with support from an information specialist based on previous syntheses[5 12] and discussion among the review team. In particular, a synthesis on patient factors affecting recruitment[20] which was ongoing at the time of compiling this review, helped to inform the search strategy. The methods set out in table 1 informed how the search strategy was structured.

The forthcoming review will employ a systematic search of the online literature, focusing on electronic databases. Databases to be searched will include Ovid MEDLINE, CINAHL, EMBASE, PsycInfo, Cochrane Central Register of Controlled Trials, ORRCA and Web of Science. A copy of the MEDLINE search strategy is shown in online supplemental appendix 1 (QES OvidMEDLINE Search Strategy). This search strategy will be adapted for all other database searches.

A search of 'grey' literature (eg, theses) will be undertaken, using tools such as OpenGrey, ProQuest and Ethos. Searching for 'grey' literature is known to be challenging[18] and potentially unsystematic, but it was agreed that a search would still add value to the review. References of all studies included in the review will be searched, and citation searches using Scopus, Web of Science and Google Scholar will also be conducted.

Due to limitations in time and resource, only literature reported in English will be included. No limits in terms of geography or time will be applied at the initial search stage.

### Selection of studies
All references identified from the search will be imported into EndNote X9. Duplicates will be removed by NF. Publications will then be imported into Microsoft Excel, where screening of titles and abstracts will be completed. A random sample of 10% of titles and abstracts will be independently screened by one other member of the review team. On completion of initial screening, full-text articles will be retrieved for all agreed articles. The primary reviewer will review all full texts and a second reviewer will full text screen 5% of the retrieved articles. Reasons for exclusion according to the eligibility criteria will be documented from this stage to inform the Preferred Reporting Items for Systematic Reviews and Meta-Analyses (PRISMA) diagram. If agreement between the primary screeners cannot be reached after discussion, a third member of the review team will be consulted to review the full text for each study where there is disagreement. If the review team have concerns about the number of discrepancies between the first and second reviewers, the reasons underpinning discrepancies will be discussed and resolved, and the second reviewer will review a further sample of papers.

| Table 1 | SPIDER search tool |
|---|---|
| Sample (S) | Participants will include recruiters with a reported role in approaching potential participants (eg, patients, carers or parents) to take part in a healthcare related RCTs. Recruiters with a range of professional roles will be included, irrespective of whether they are registered professionals. Examples of professions with recruiter roles include: doctors, surgeons, physiotherapists, nurses, radiographers, GPs, clinical trials assistants, research practitioners. |
| Phenomena of interest (Pi) | The phenomena of interest in this study is recruitment to RCTs. Studies which consider recruitment alongside other trial related activities may be included as long as the recruitment element is clearly reported and distinguishable from other trial related activities, in so far as findings can be extracted for inclusion in analysis. |
| Design (D) | Primary research that uses qualitative approaches/designs to investigate recruiters' views, experiences and practices/behaviour related to attempts to recruit participants into RCTs will be considered for inclusion. No limits on qualitative theoretical frameworks will be applied. Qualitative methods of data collection will include, but not be limited to, qualitative interviews (in-depth, unstructured, semi-structured and structured), focus groups and observations (participant/non-participant). The review will focus on studies that report on recruitment to particular types of RCTs: those based in the healthcare sector, and those that randomise at the individual patient (or proxy) level. Research reporting on cluster randomised trials will not be included because they are likely to face distinct challenges and may not follow the same process of recruitment as RCTs that randomise at the individual level. Studies that report results from non-human, non-healthcare or laboratory-based RCTs will be excluded. The definition of RCTs may include pilot or feasibility studies, provided that they are randomised. Qualitative studies of hypothetical RCTs will be excluded. |
| Evaluation (E) | This synthesis will explore the attitudes, experiences and practices/behaviours of those recruiting patients into RCTs. |
| Research type (R) | The search will focus on qualitative research, although-mixed methods research will be considered for inclusion where the qualitative element is clearly defined and reported. |
| Comparisons | It is likely that there will be primary reports from a range of RCTs. We will explore comparisons across specialty or clinical field (eg, oncology and radiology), recruiter's professional role, level of care (eg, primary, secondary, community), nature of RCT treatment arms (eg, standard care vs novel treatment or less/no treatment), and any other factors of perceived importance that emerge. These comparisons may allow for a greater level of examination of some of the intricacies of recruiting to RCTs and may yield insights that are specific to particular contexts or therapeutic areas. |

GPs, general practitioners; RCTs, randomised controlled trials; SPIDER, Sample, Phenomenon of Interest, Design, Evaluation, Research type.

## Sampling of studies

Initial scoping work has indicated that a potentially large number of studies (>100) will meet the inclusion criteria and therefore be potentially relevant to the evidence synthesis. Once duplicates have been removed and full-text screening has taken place, the review team will discuss the manageability of the volume of eligible studies. The information collected (outlined in table 2) will provide insight into the range and volume of literature. If the body of research potentially eligible for inclusion is determined by the review team to be so great as to potentially undermine the depth of the review, a sampling strategy will be applied, based on the amount and range of studies eligible for inclusion.[21 22] For example, we will consider using a purposeful sampling approach, which has been advocated as a method of reducing the number of studies involved in syntheses without compromising the integrity of the analysis.[22] This method is endorsed by Suri[23] in their overview of methods of sampling for QES, and several practical examples describing and explaining the use of purposeful sampling in qualitative syntheses have

been published.[22 24] If applied, specifics of this method of sampling will be discussed and documented by the review team but are likely to include criteria such as maximum variation of disease areas, patient age groups and types of intervention. An alternate approach would be the application of the CART framework (Completeness, Accuracy, Relevance and Timeliness): a method of assessing the evidence for inclusion in qualitative reviews.[25] Using the CART framework would help to ensure that the included research is closely linked to the synthesis question. If used, the definitions for each of the terms within the CART framework will be developed by the review team and agreed before being applied, but will be based on those definitions used in the INTERUPT systematic review, which were published as part of their CART protocol.[26] If this approach is used, two members of the review team will apply the agreed criteria to the studies following the piloting of the criteria on a random subset of studies. A decision around the method of sampling will be made following full text screening, once the review team have considered the scope and breadth of included articles in

| Table 2 | Proposed contextual details data extraction fields | | |
|---|---|---|---|
| **Study characteristics** | **Qualitative characteristics** | **RCT characteristics** | **Participant (recruiter) characteristics** |
| ► Author<br>► Journal<br>► Date of study<br>► Country of study | ► Data collection method<br>► Data analysis method<br>► Sample size | ► Healthcare setting<br>► Disease area<br>► Intervention<br>► Comparators | ► Profession<br>► Experience of recruiters (as reported in study) |

RCT, randomised controlled trials.

relation to the criteria specified in table 2. The choice of sampling strategy will be primarily driven by the team's judgement about which method is most likely to yield a final sample of articles that will comprehensively address the review question.

## Data extraction and management

QSR NVivo will be used to manage extraction of findings and key contextual details will be reported in Microsoft Excel. Initially, key contextual details will be extracted from all articles deemed eligible from full-text screening. Examples of the types of data to be extracted from individual studies are shown in table 2. The proposed data extraction fields will be piloted with a small number of studies and discussed among the review team to ensure appropriate information is captured.

Full data extraction (eg, findings) will be completed following the application of a sampling strategy. As qualitative findings can be reported outside of the traditional results headings, extraction will not be limited to any specific heading within the articles. A second member of the review team will extract data for 10% of the studies.

## Assessment of methodological limitations in primary studies

The Critical Appraisal Skills Programme (CASP)[27] offers a tool with which to appraise qualitative research and is often used in QES.[20] The CASP is adaptable and can be amended to emphasise subsections which are particularly pertinent to the research question. The use of the CASP is also recommended by Cochrane and feeds into the Confidence in Evidence from Reviews of Qualitative Research (GRADE-CERQual) process.[28] As such, the CASP checklist may be modified to enable a more comprehensive appraisal of study conduct and reporting. The purpose of applying the CASP tool will not be to exclude studies, rather to assess the methodological strengths and weaknesses of each study. This is in line with the methodological debate over whether it is in keeping with a qualitative methodology to make assessments of quality and to exclude studies on this basis.[29 30]

Two members of the review team will use the tool to assess the methodological limitations of each study. A third member will be consulted in the case of disagreements.

## Data synthesis

A thematic synthesis approach, as outlined by Thomas and Harden[31] will be used to analyse and synthesise the extracted data. The INTEGRATE-HTA guidance[32] in selecting a review methodology was followed and confirmed the appropriateness of undertaking a thematic synthesis. This guidance encourages the reviewers to select the appropriate review methodology by outlining areas of reflection and consideration. As such, the review team considered the anticipated volume of studies to review (based off initial scoping work) and the intended audience of the review, namely academics, clinical professionals and professionals involved in trial conduct, and concluded that thematic synthesis would be

an appropriate approach. Following a thematic synthesis approach will also allow this review to be aligned with the synthesis produced from the Houghton *et al*[20] protocol.

A three-stage thematic synthesis as outlined by Thomas and Harden[31] will be undertaken, consisting of the following steps:

1. Line by line coding of the 'results' (not limited to the results section) of the study reports. Codes will be attached to sentences or quotes to allow the translation of concepts between the studies.[33] Codes will continuously be checked for consistency and where appropriate very similar codes may be consolidated.
2. Development of descriptive themes encompassing a range of similar codes.
3. Development of analytical themes: ideas and constructs beyond what are directly described in the data will be developed iteratively. These analytical themes will be checked and cross checked with the original texts, codes and descriptive themes. Once analytical themes have been developed, interpretations will be discussed within the team and related back to the question addressed by the review.

## Assessment of confidence in the review findings

In order to be able to assess confidence in the review findings, the GRADE-CERQual approach, developed and outlined by Lewin *et al*[34] will be used. This is in keeping with Houghton *et al*'s[20] review on patient perspectives and its use is endorsed by the Cochrane Qualitative and Implementation Methods Group.[28]

The GRADE-CERQual approach focuses on four key components: methodological limitations, coherence, adequacy of data and relevance. Assessment of methodological limitations will be conducted using the CASP, as previously described. Coherence will be addressed by reviewers being reflexive when determining themes and concluding findings, ensuring that results are presented in a way that shows a strong grounding in the primary data. In order to address adequacy of the data in the synthesis, a sensitivity analysis will be conducted where studies that are deemed to be of particularly high or low overall quality will be removed. This will allow observation of whether any individual studies make an overall difference on the findings of the synthesis.[35] Each study included in the review will be assigned a relevance category as defined by Noyes *et al*[36] as 'direct relevance', 'indirect relevance', 'partial relevance' and 'uncertain relevance'. The GRADE-CERQual assessment for each finding will be made by NF and discussed among the review authors, before being presented in a summary of qualitative findings table and full evidence profile.[37]

## Author reflexivity

The review team comprises both experienced qualitative and mixed-methods/applied researchers, some of whom have clinical backgrounds. We acknowledge that our previous research experiences may influence our choice of review methods and how we interpret the data. As such,

each team member will remain mindful of how their personal lens may influence the review process, and decisions will be made via group discussions so as to include review authors with a range of different perspectives.

### Patient and public involvement

Patients and members of the public were not involved in the design of this protocol.

## ETHICS AND DISSEMINATION

No ethical approval is required for the proposed synthesis as it will use only published literature. We will seek to publish the results of the review and both the protocol and the results will be publicised using social media. The results will supplement the 'Trinity' package of systematic reviews to systematically collate what is known about the recruitment process and inform future interventions to improve recruitment to RCTs. The review will also form a chapter of NF's PhD thesis.

**Acknowledgements** The authors thank Alison Richards for her assistance in developing the search strategy.

**Contributors** NF conceived the review idea, developed and refined the review question, contributed towards the development of the protocol and wrote the first draft of the protocol. DE developed and refined the review question, contributed towards the development of the protocol and critically reviewed and commented on drafts of the protocol. MJ conceived the review idea, developed and refined the review question, contributed towards the development of the protocol and critically reviewed and commented on drafts of the protocol. CH conceived of the review idea, contributed to the development of the protocol and critically reviewed and commented on drafts of the protocol. BY critically reviewed and commented on drafts of the protocol. JD critically reviewed and commented on drafts of the protocol. LR conceived the review idea, developed and refined the review question, contributed towards the development of the protocol and critically reviewed and commented on drafts of the protocol.

**Funding** NF's PhD work was funded by MRC Network of Hubs for Trials Methodology Research (MR/L004933/2/R9). This work was undertaken with the support of the MRC ConDuCT-II Hub (Collaboration and innovation for Difficult and Complex randomised controlled Trials In Invasive procedures-MR/K025643/1). DE was supported by the NIHR Biomedical Research Centre at University Hospitals Bristol NHS Foundation Trust and the University of Bristol (reference number BRC-1215-20011).

**Competing interests** None declared.

**Patient consent for publication** Not required.

**Provenance and peer review** Not commissioned; externally peer reviewed.

**ORCID iDs**
Nicola Farrar http://orcid.org/0000-0002-9473-5678
Daisy Elliott http://orcid.org/0000-0001-8143-9549
Marcus Jepson http://orcid.org/0000-0003-3261-1626
Catherine Houghton http://orcid.org/0000-0003-3740-1564
Bridget Young http://orcid.org/0000-0001-6041-9901
Jenny Donovan http://orcid.org/0000-0002-6488-5472
Leila Rooshenas http://orcid.org/0000-0002-6166-6055

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
