## [Reviewer comments · BMJ Open]

ARTICLE DETAILS

TITLE (PROVISIONAL)	Recruiters' perspectives and experiences of trial recruitment processes: a qualitative evidence synthesis protocol
AUTHORS	Farrar, Nicola; Elliott, Daisy; Jepson, Marcus; Houghton, Catherine; Young, Bridget; Donovan, Jenny; Rooshenas, Leila

VERSION 1 – REVIEW

REVIEWER	Taylor, Rachel University College London Hospitals NHS Foundation Trust, CNMAR
REVIEW RETURNED	27-Jan-2021

GENERAL COMMENTS	Thank you for asking me to review this protocol for a review of healthcare professional experience of recruiting patients to clinical trials. In its current form I do not think this is suitable for publication in BMJ Open. I think the major criticism of this review is the type of review is not clearly articulated so the authors seem to have used elements of various types – in Prospero it is listed as a systematic review, yet the authors state they will not be including all the evidence identified. Some parts seem like a scoping review, while other sections read as a narrative review. A number of other comments the authors might want to consider to strengthen the reporting of this protocol: • CIHAHL is listed as one of the databases being searched. I assume this is the Cumulative Index to Nursing and Allied Health Literature (CINAHL)? If so, it is consistently spelt incorrectly throughout the protocol.• The authors refer to clinical and non-clinical. I would suggest using different terminology –clinical trials assistant and clinical trials practitioner are clinical (the hint is in the job title). If the authors mean registered professional and non-registered professional then they should use these terms.• It is unclear why cluster randomised controlled trials are excluded because the process of recruitment is the same. I also wonder why other trial designs are not included, for example step wedge designs, waiting list.• It needs to specify the type of intervention – are these CTIMP trials or are complex interventions also included. Specifying the type of intervention may reduce the number of papers included and provide a more valid way of including evidence other than purposive sampling.• In addition to the above, what phase of study is going to be included? The process of recruitment and decision-making in a phase I oncology trial, for example, is going to be quite different to a phase III/IV trial. Again, having more specific inclusion/exclusion criteria will enable the number of papers to be limited in a more
--

	robust way that does not introduce the bias through a purposive inclusion method.  • As a protocol it should be possible to replicate this review but some of the detail provided is not precise; more detail is needed. For example, which data analysis package are publications being imported into, a 'proportion' of the process will be reviewed – what proportion, and is there any evidence to support this? • If there is not agreement between primary screeners should they not be reviewing more than 10%? This suggests there could be errors elsewhere. • It states on Prospero that this is a systematic review, which would require the inclusion of all the identified evidence but the authors note the search will identify a lot of relevant publications so they will purposively select what to include. There is a lot of potential for bias in doing this, which needs to be acknowledged and discussed. • The criteria for purposive sampling needs to be specified. • There are a lot of frameworks being used in this review (CASP CART Framework, INTEGRATE_HTA guidance, CERQual assessment). Are they all necessary? Seems to be more than any other review I have read and how is this going to influence the interpretation of the synthesis of the evidence? • Data extraction: this is missing participant characteristics of the qualitative study not the RCT and it will be important to record the type of qualitative study. • The authors are using CASP to assess quality, which is quite a superficial method of assessment. Qualitative studies submitted to BMJ Open are recommended to include a COREQ assessment, which will pick up limitations of bias not captured by CASP. • If the authors need additional criteria to include full papers, rather than the introduction of bias through purposive sampling then excluding according to quality would be a logical solution. Other sampling strategies would mean the authors could ignore or miss a lot of good quality evidence while including a lot of poor evidence.
--	---

REVIEWER	Grindle, Mark University of the Highlands and Islands, Institute for Health Research and Innovation
REVIEW RETURNED	25-Feb-2021

GENERAL COMMENTS	This is a well-written protocol and timely - As the author recognises, recruitment to RCT's is increasingly problematic and costly: A more thorough qualitative understanding of the barriers to take-up is much needed. A tiny point...13 - 'A synthesis of patients' perspectives is soon to report'. could be more clearly phrased (?)
---

VERSION 1 – AUTHOR RESPONSE

Reviewer 1

Thank you for asking me to review this protocol for a review of healthcare professional experience of recruiting patients to clinical trials. In its current form I do not think this is suitable for publication in BMJ Open. I think the major criticism of this review is the type of review is not clearly articulated so the authors seem to have used elements of various types – in Prospero it is listed as a systematic

review, yet the authors state they will not be including all the evidence identified. Some parts seem like a scoping review, while other sections read as a narrative review.

Thank you very much for the time you have taken to review the protocol. We believe we have addressed all of your comments, as outlined below. We feel the protocol is now much improved following the additions and clarifications.

We hope we have now fully clarified the type of review. We have more clearly articulated that the proposed review is a qualitative evidence synthesis (QES), as shown in the revised sections of 'Objectives' (page 4) and 'Methods and analysis' (page 4). The development of this protocol was informed by literature relevant to the design and conduct of evidence synthesis,¹⁻⁵ which acknowledges that this is a relatively new and developing field.⁶⁻⁸ This protocol proposes a review that will use established methods of sampling, analysis and assessments of confidence in the findings, in keeping with a qualitative evidence synthesis.

Thank you for pointing the inaccurate PROSPERO entry out; as you have correctly noted, this required updating to reflect the progress made on this synthesis and has now been updated. The PROSPERO entry was set-up prior to the protocol being finalised and as such several 'types and methods' were selected. This has now been updated to just reflect that it is a 'synthesis of qualitative studies', as specified in the protocol. Unfortunately it has not been possible to as yet remove 'systematic review' from the 'type and method of review' section of the PROSPERO entry, but the review team have contacted PROSPERO directly to amend this.

A number of other comments the authors might want to consider to strengthen the reporting of this protocol:

- CIHAHL is listed as one of the databases being searched. I assume this is the Cumulative Index to Nursing and Allied Health Literature (CINAHL)? If so, it is consistently spelt incorrectly throughout the protocol.

Thank you for pointing this out. This was a typo and has now been correct to CINAHL throughout.

- The authors refer to clinical and non-clinical. I would suggest using different terminology –clinical trials assistant and clinical trials practitioner are clinical (the hint is in the job title). If the authors mean registered professional and non-registered professional then they should use these terms.

Thank you for raising this important point. We agree that stating roles may be 'clinical' or 'non-clinical' does not accurately distinguish between the range of roles that can take on the responsibility of recruiting to RCTs. As such, we have made several changes to the SPIDER search tool (Table 1, pages 4/5), in particular to the 'Sample' and 'Comparison' headings. In 'Sample', we have now stated: "Recruiters with a range of professional roles will be included, irrespective of whether they are registered professionals. Examples of professions with recruiter roles include. doctors, surgeons, physiotherapists, nurses, radiographers, GPs, clinical trials assistants, research practitioners."

In the 'Comparisons' section, we now state:

"We will explore comparisons across specialty or clinical field (e.g. oncology and radiology), recruiter's professional role, level of care (e.g. primary, secondary, community), nature of RCT treatment arms (e.g. standard care vs novel treatment, or less/no treatment), and any other factors of perceived importance that emerge."

- It is unclear why cluster randomised controlled trials are excluded because the process of recruitment is the same. I also wonder why other trial designs are not included, for example step wedge designs, waiting list.

Cluster RCTs have been excluded due to differences in recruitment processes relative to individual-level randomized controlled trials. We have updated the SPIDER search tool (Table 1, pages 4/5, section 'Design') to more clearly reflect our reasoning for excluding cluster RCTs. Whilst the review team acknowledge that the rest of the review question is broad, this was one area where we agreed in advance that the 'type' of RCT would not include different design types. We agree that this may be a limitation of the review and will comment upon this when discussing the findings of the synthesis. A separate evidence synthesis exploring the views of recruiters of such types of trials may be warranted and could form an interesting comparison to the results of this review.

- It needs to specify the type of intervention – are these CTIMP trials or are complex interventions also included. Specifying the type of intervention may reduce the number of papers included and provide a more valid way of including evidence other than purposive sampling.

This is an interesting point; there may be potential to conduct more specific qualitative evidence syntheses of particular types of trial (e.g. those with a surgical intervention), but the aim of this review was broader given the amount of time that has passed since the last review using systematic methods was published.⁹ The need for this broad review of recruiters' perspectives has also been identified as a priority for future research in a recent Cochrane evidence synthesis.¹⁰ We have now highlighted this in the updated introduction section (page 3) of the protocol:

“An up-to-date evidence synthesis of recruiters' (that encompasses all professional roles) perspectives and experiences of recruiting to RCTs (specifically) is now warranted,¹⁰ given the limitations of previous reviews of recruiter perspectives, and the growth of interest on this topic since the last search (2013) was conducted.”

As such, we will not place limits on the types of trial recruiters work on, but will record this information for contextual purposes. We agree with the reviewer that future, more niche reviews of recruiter perspectives on working on particular types of trial may be warranted – something that may become clearer once the findings of our proposed review are reported.

We address the reviewer's concerns about purposive sampling below.

- In addition to the above, what phase of study is going to be included? The process of recruitment and decision-making in a phase I oncology trial, for example, is going to be quite different to a phase III/IV trial. Again, having more specific inclusion/exclusion criteria will enable the number of papers to be limited in a more robust way that does not introduce the bias through a purposive inclusion method.

Thank you for raising this for consideration. We agree that the recruitment process may be different between very early and late phase trials – something that, to our knowledge, has not yet been empirically examined in detail. This is a valuable point that we will certainly consider when reflecting on the findings and implications for future research that stem from the current proposed review.

- As a protocol it should be possible to replicate this review but some of the detail provided is not precise; more detail is needed. For example, which data analysis packages are publications being imported into, a 'proportion' of the process will be reviewed – what proportion, and is there any evidence to support this?

Thank you for pointing out these details were missing; they have now been included in the protocol as requested, in particular in the sections 'Selection of studies' (page 6) and 'Data extraction and management' (page 7/8). We have now explained that publications will be screened in Microsoft Excel and the selected studies will be imported into NVivo for synthesis. A second reviewer will undertake full text screening of 5% of the papers that are considered potentially eligible from the title and abstract screening stage. The same reviewer will extract data from 10% of the studies included for analysis. To our knowledge, there is no evidence to support what percentage of the review process should be done by a second team member for qualitative evidence syntheses; as such we have been guided by the time and resources available for the proposed review.

- If there is not agreement between primary screeners should they not be reviewing more than 10%? This suggests there could be errors elsewhere.

Considerable thought was given by the review team as to the percentage of double screening/extraction that should be done by a second reviewer. As mentioned above, there is little guidance to drive this decision, and the team are limited by the time and resource available. We have now clarified the steps that will be taken if there is disagreement, and added extra detail to explain that further double screening will occur if deemed necessary following identification of discrepancies and discussion with a third reviewer (Selection of studies, page 6):

“If agreement between the primary screeners cannot be reached after discussion, a third member of the review team will be consulted to review the full text for each study where there is disagreement. If the review team have concerns about the number of discrepancies between the first and second

reviewers, the reasons underpinning discrepancies will be discussed and resolved, and the second reviewer will review a further sample of papers.”

- It states on Prospero that this is a systematic review, which would require the inclusion of all the identified evidence but the authors note the search will identify a lot of relevant publications so they will purposively select what to include. There is a lot of potential for bias in doing this, which needs to be acknowledged and discussed.

Thank you for raising this. The review team has contacted PROSPERO about removing the ‘systematic review’ entry in the PROSPERO form, and this will be rectified as soon as possible. We acknowledge the reviewer’s concern that purposive sampling can introduce bias. Sampling purposively is well established in qualitative data collection and analysis, with qualitative researchers often identifying and using ‘information rich’ cases where a breadth of understanding can be obtained.¹¹ As with sampling in primary qualitative research, those undertaking qualitative evidence synthesis must ensure that too much data is not included in the review, so as to risk undermining the reviewers’ abilities to thoroughly engage with the data.² Purposive sampling offers a robust and commonly used method of achieving such a sample.² Suri highlights how the method of sampling must be aligned to the purpose of the synthesis, and that different types of purposeful sampling can be appropriate for different approaches to qualitative evidence synthesis.¹² Suri’s descriptions of the different types of purposeful sampling would be used to guide decision making about the specific type of sampling (e.g. maximum variation, criterion).

- The criteria for purposive sampling needs to be specified.

The sampling criteria will be informed by the volume and variety of studies which are considered eligible for inclusion after full text screening. A sampling framework will be developed once the key contextual information of eligible papers has been extracted (as described in Table 2), which will help to guide the sampling strategy. Detailed records of the process will be kept, and a full and transparent account of the sampling strategy adopted will be detailed in the ‘Methods’ that will accompany the review findings. Developing a sampling framework unique to each QES review following full text screening has been reported in the literature.² We will ensure that the sampling strategy is informed by established and transparent processes, for example, we envisage that purposive sampling (if used) would be guided by the strategies proposed by Suri et al.¹² We have clarified how the type of sampling will be decided upon in ‘sampling of studies’ (page 8):

“A decision around the method of sampling will be made following full text screening, once the review team have considered the scope and breadth of included articles in relation to the criteria specified in Table 2. The choice of sampling strategy will be primarily driven by the team’s judgment about which method is most likely to yield a final sample of articles that will comprehensively address the review question.”

- There are a lot of frameworks being used in this review (CASP CART Framework, INTEGRATE_HTA guidance, CERQual assessment). Are they all necessary? Seems to be more than any other review I have read and how is this going to influence the interpretation of the synthesis of the evidence?

We are happy to clarify the value and need of each of these frameworks, which each serve distinct purposes. The CASP (an assessment of methodological limitations of primary research) will be used to make assessments of quality of the included studies. This is part of the GRADE CERQual assessment, which was designed to assess the degree of confidence that can be placed in the findings of a review.³ The use of GRADE CERQual is recommended by Cochrane to assess confidence in review findings.¹³ The CART criteria are one method of sampling studies which have been used in other reviews.^{10 14} As indicated in the protocol, however, the method of sampling will be determined when the authors have a better sense of the number of studies which are potentially eligible for inclusion.

The INTEGRATE HTA guidance was followed when determining what type of qualitative evidence synthesis should be undertaken. As there are several different methodologies for synthesising qualitative data (e.g. meta-ethnography, narrative synthesis), the authors wanted to

make sure that the methodology used for this synthesis was appropriate for the review question, experience of the team, timeframe and other criteria suggested by the INTEGRATE guidance as important to consider.⁴

- Data extraction: this is missing participant characteristics of the qualitative study not the RCT and it will be important to record the type of qualitative study.

We fully agree that this additional information is helpful. Table 2 (page 7) has now been updated to clarify that we will be extracting details about the participants (recruiters) as well as details about the RCT. As described in Table 2 we will initially report the recruiters' profession and their experience of recruiting (however this is defined in the reporting study).

- The authors are using CASP to assess quality, which is quite a superficial method of assessment. Qualitative studies submitted to BMJ Open are recommended to include a COREQ assessment, which will pick up limitations of bias not captured by CASP.

We agree with the reviewer that CASP does have its limitations. We chose to use this quality assessment tool because it is used as part of the wider GRADE CERQual assessment of confidence in the qualitative findings. The use of CASP for assessing quality in qualitative evidence synthesis is endorsed by Cochrane,¹³ and also aligns this review with a previously published Cochrane review of patient perspectives and experiences of recruitment.¹⁰ The review team may adapt the CASP if there are deemed to be limitations that restrict the assessment of included studies, as has been done by others who have used CASP.¹⁵ If the review team does amend the CASP, any modifications will be reported and justified in the synthesis. If the CASP is used unmodified, particular attention will be paid to addressing the 'hints' listed alongside the CASP questions, which add depth to assessments.

- If the authors need additional criteria to include full papers, rather than the introduction of bias through purposive sampling then excluding according to quality would be a logical solution. Other sampling strategies would mean the authors could ignore or miss a lot of good quality evidence while including a lot of poor evidence.

Thank you for your suggestion. We hope our expansion on sampling and explanations above now provide a clearer account of why we have included purposive sampling as a potential approach for this qualitative evidence synthesis. We gave much thought to the possibility of using CASP to inform sampling decisions, and consulted the existing methodological literature and previously published qualitative evidence syntheses to inform our approach. We acknowledge that in some review types, particularly quantitative reviews that focus on the effectiveness of interventions, decisions around sampling are often made with the risk of bias in mind. There is, however, considerable methodological debate over whether it is in keeping with a qualitative methodology to exclude studies on the basis of quality assessments.^{16 17} Cochrane recommend using quality appraisal as a method of assessing the methodological limitations of the review findings, something that they consider essential for reviews.¹³ In keeping with this, we intend to use the CASP tool to assess and appraise the methodological limitations of the primary studies as endorsed by Cochrane,¹⁷ rather than as a tool to exclude studies. We will continue to follow methodological developments and debate in this area and reflect on this when reporting the findings of the intended synthesis.

Reviewer 2

This is a well-written protocol and timely - As the author recognises, recruitment to RCT's is increasingly problematic and costly: A more thorough qualitative understanding of the barriers to take-up is much needed.

A tiny point...¹³ - 'A synthesis of patients' perspectives is soon to report'. could be more clearly phrased (?)

Thank you for taking the time to review the protocol and for your positive feedback. We're pleased to read that the protocol was judged to be well-written and timely.

Thank you for your helpful comment about rephrasing this sentence. Given the synthesis of patients' perspectives has now been published, this statement has been replaced with:
"The literature on patient related factors is well developed, with a Cochrane evidence synthesis recently reporting its findings.¹⁰ (Introduction, page 2)"

References

1. Harris JL, Booth A, Cargo M, et al. Cochrane Qualitative and Implementation Methods Group guidance series—paper 2: methods for question formulation, searching, and protocol development for qualitative evidence synthesis. *J Clin Epidemiol* 2018;97:39-48. doi: 10.1016/j.jclinepi.2017.10.023
2. Ames H, Glenton C, Lewin S. Purposive sampling in a qualitative evidence synthesis: a worked example from a synthesis on parental perceptions of vaccination communication. *Bmc Med Res Methodol* 2019;19(1):26. doi: 10.1186/s12874-019-0665-4
3. Lewin S, Booth A, Glenton C, et al. Applying GRADE-CERQual to qualitative evidence synthesis findings: introduction to the series. *Implement Sci* 2018;13(Suppl 1):2. doi: 10.1186/s13012-017-0688-3 [published Online First: 2018/02/01]
4. Booth A, Noyes J, Flemming K, et al. Guidance on choosing qualitative evidence synthesis methods for use in health technology assessments of complex interventions, 2016.
5. Booth A, Noyes J, Flemming K, et al. Structured methodology review identified seven (RETREAT) criteria for selecting qualitative evidence synthesis approaches. *J Clin Epidemiol* 2018;99:41-52. doi: 10.1016/j.jclinepi.2018.03.003 [published Online First: 2018/03/20]
6. Flemming K, Booth A, Garside R, et al. Qualitative evidence synthesis for complex interventions and guideline development: clarification of the purpose, designs and relevant methods. *BMJ Global Health* 2019;4(Suppl 1):e000882. doi: 10.1136/bmjgh-2018-000882
7. Flemming K, Noyes J. Qualitative Evidence Synthesis: Where Are We at? *International Journal of Qualitative Methods* 2021;20:1609406921993276. doi: 10.1177/1609406921993276
8. Noyes J, Booth A, Cargo M, et al. Cochrane Qualitative and Implementation Methods Group guidance series—paper 1: introduction. *J Clin Epidemiol* 2018;97:35-38. doi: <https://doi.org/10.1016/j.jclinepi.2017.09.025>
9. Newington L, Metcalfe A. Researchers' and clinicians' perceptions of recruiting participants to clinical research: a thematic meta-synthesis. *J Clin Med Res* 2014;6(3):162-72. doi: 10.14740/jocmr1619w [published Online First: 2014/04/16]
10. Houghton C, Dowling M, Meskell P, et al. Factors that impact on recruitment to randomised trials in health care: a qualitative evidence synthesis. *Cochrane Db Syst Rev* 2020(10) doi: 10.1002/14651858.MR000045.pub2
11. Palinkas LA, Horwitz SM, Green CA, et al. Purposeful Sampling for Qualitative Data Collection and Analysis in Mixed Method Implementation Research. *Adm Policy Ment Health* 2015;42(5):533-44. doi: 10.1007/s10488-013-0528-y [published Online First: 2013/11/07]
12. Suri H. Purposeful Sampling in Qualitative Research Synthesis. *Qualitative Research Journal* 2011;11(2):63-75. doi: 10.3316/QRJ1102063
13. Noyes J, Booth A, Flemming K, et al. Cochrane Qualitative and Implementation Methods Group guidance series—paper 3: methods for assessing methodological limitations, data extraction and synthesis, and confidence in synthesized qualitative findings. *J Clin Epidemiol* 2018;97:49-58. doi: <https://doi.org/10.1016/j.jclinepi.2017.06.020>
14. Whitaker R, Hendry M, Aslam R, et al. Intervention Now to Eliminate Repeat Unintended Pregnancy in Teenagers (INTERUPT): a systematic review of intervention effectiveness and cost-effectiveness, and qualitative and realist synthesis of implementation factors and user engagement. *Health Technol Assess* 2016;20(16):1-214. doi: 10.3310/hta20160 [published Online First: 2016/03/05]
15. Atkins S, Lewin S, Smith H, et al. Conducting a meta-ethnography of qualitative literature: Lessons learnt. *Bmc Med Res Methodol* 2008;8(1):21. doi: 10.1186/1471-2288-8-21

16. Campbell R, Pound P, Pope C, et al. Evaluating meta-ethnography: a synthesis of qualitative research on lay experiences of diabetes and diabetes care. Soc Sci Med 2003;56(4):671-84.

doi: [https://doi.org/10.1016/S0277-9536\(02\)00064-3](https://doi.org/10.1016/S0277-9536(02)00064-3)

17. Long HA, French DP, Brooks JM. Optimising the value of the critical appraisal skills programme (CASP) tool for quality appraisal in qualitative evidence synthesis. Research Methods in Medicine & Health Sciences 2020:2632084320947559. doi: 10.1177/2632084320947559

VERSION 2 – REVIEW

REVIEWER	Taylor, Rachel University College London Hospitals NHS Foundation Trust, CNMAR
REVIEW RETURNED	14-Jun-2021

GENERAL COMMENTS	Thank you for asking me to re-review this manuscript and thank you to the authors for addressing all my comments. I am happy that these have been all thoroughly addressed and I look forward to reading the finished review.
---